# The Landscape of Telomere Length and Telomerase in Human Embryos at Blastocyst Stage

**DOI:** 10.3390/genes14061200

**Published:** 2023-05-30

**Authors:** Fang Wang, David H. McCulloh, Kasey Chan, Ashley Wiltshire, Caroline McCaffrey, James A. Grifo, David L. Keefe

**Affiliations:** 1NYU Langone Fertility Center, New York, NY 10022, USA; david.mcculloh@nyulangone.org (D.H.M.); ashleyfimrc@gmail.com (A.W.); caroline.mccaffrey@nyulangone.org (C.M.); james.grifo@nyulangone.org (J.A.G.); david.keefe@nyulanong.org (D.L.K.); 2Department of Obstetrics and Gynecology, NYU Grossman, School of Medicine, New York, NY 10016, USA

**Keywords:** telomere length, telomerase activity, human embryo, aneuploidy, blastocyst

## Abstract

The telomere length of human blastocysts exceeds that of oocytes and telomerase activity increases after zygotic activation, peaking at the blastocyst stage. Yet, it is unknown whether aneuploid human embryos at the blastocyst stage exhibit a different profile of telomere length, telomerase gene expression, and telomerase activity compared to euploid embryos. In present study, 154 cryopreserved human blastocysts, donated by consenting patients, were thawed and assayed for telomere length, telomerase gene expression, and telomerase activity using real-time PCR (qPCR) and immunofluorescence (IF) staining. Aneuploid blastocysts showed longer telomeres, higher telomerase reverse transcriptase (TERT) mRNA expression, and lower telomerase activity compared to euploid blastocysts. The TERT protein was found in all tested embryos via IF staining with anti-hTERT antibody, regardless of ploidy status. Moreover, telomere length or telomerase gene expression did not differ in aneuploid blastocysts between chromosomal gain or loss. Our data demonstrate that telomerase is activated and telomeres are maintained in all human blastocyst stage embryos. The robust telomerase gene expression and telomere maintenance, even in aneuploid human blastocysts, may explain why extended in vitro culture alone is insufficient to cull out aneuploid embryos during in vitro fertilization.

## 1. Introduction

Telomeres, the unique DNA–protein complexes with variable numbers of telomeric DNA repeats (TTAGGG), protect chromosome ends from DNA degradation, the DNA damage response, and chromosome fusions [1]. Telomere length is primarily maintained by telomerase, a reverse transcriptase composed of a catalytic unit (telomerase reverse transcriptase, TERT) and an RNA template (telomerase RNA component, TERC) [2]. In addition, homology-directed telomere synthesis, called alternative lengthening of telomere (ALT), can maintain telomere length when telomerase activity is inefficient or absent [3]. Cancer cells escape senescence by maintaining telomere length via telomerase or ALT [4]. In contrast, telomeres in somatic cells, which lack telomerase or ALT, progressively shorten as cells divide [5].

Telomerase activity in mouse, bovine, and human oocytes is low or undetectable, and telomeres are remarkably short in this differentiated, aging-sensitive cell type [6]. In contrast, telomerase activity in spermatogonia is high, so telomeres in sperm are much longer than in oocytes or somatic cells [7]. After zygotic genome activation, telomerase activity gradually increases and peaks at blastocyst stage. Telomeres lengthen from zygote to blastocyst stages, as determined by both qPCR and Quantitiative Fluorescent in situ hybridization (Q-FISH) [8]. Telomerase repeated amplification protocol (TRAP) assay demonstrates very low telomerase activity in human oocytes and cleavage stage embryos, with the highest level of telomerase activity being at the blastocyst stage of development [9]. Telomere dynamics, telomerase gene expression, and activity in human blastocysts remain poorly studied.

In most cell types, the loss of telomeric DNA repeats or deficiencies in telomeric proteins lead to chromosome fusions and chromosome instability [10]. Consistent with this, gametes from telomerase null mice (mTR^−/−^) develop poorly after in vitro fertilization (IVF) and lead to aberrant embryos, with impaired preimplantation development [11]. Moreover, late-generation mTR^−/−^ mice exhibit an abnormal reproductive phenotype, with severe germ cell depletion [12]. In women, telomere length has been linked to the developmental competency of early embryos. Women with advanced reproductive age have shorter leukocyte telomere length and higher rates of aneuploid embryo when undergoing IVF treatment [13,14]. Women with telomeropathies suffer decreased ovarian reserve [15] and increased risk of aneuploid embryos following IVF, with preimplantation genetic testing for aneuploidies (PGT-A) [16]. In contrast, telomere length in spermatozoa increases with paternal age, presumably due to the high telomerase activity in spermatogonia. Intriguingly, advanced paternal age also increases sperm telomere length heterogeneity, and may lower the quality of spermatozoa [17,18].

A recent study reported no difference in telomere length between euploid vs. aneuploid cleavage stage human embryos through single cell whole genome amplification (WTA) assay [19]. Telomere content measured by the qPCR of trophectoderm cells also did not differ between biopsied aneuploid and euploid blastocysts, though the polar bodies of oocytes which gave rise to aneuploid blastocysts contained decreased telomere content [20]. Mania et al. [21] found shorter telomeres in single blastomeres from aneuploid compared to euploid cleavage stage embryos. Together, these studies suggest telomere length resets from oocyte to blastocyst stage in human preimplantation embryo development, but the profiles of telomere length, telomerase gene expression, and activity in human aneuploid blastocysts remain poorly understood. In this study, we compared telomere length, telomerase gene expression, and activity between euploid and aneuploid human blastocysts.

## 2. Materials and Methods

### 2.1. Ethical Approval

Cryopreserved human blastocysts donated by patients with written informed research consent under NYU Institutional Review Board (IRB) approval (S16-00154) were de-identified after the extraction of key clinical parameters.

### 2.2. Materials

All reagents were purchased from MilliporeSigma if not otherwise stated. All DNA oligos were synthesized by Integrated DNA Technologies and dissolved in TE buffer (pH = 8) or nuclease-free water for stocking solution at 100 µM. Working solutions of oligos were diluted in nuclease-free water to obtain the required concentration.

Donated blastocysts were selected for the study according to results of their PGT-A analysis. The ploidy status of each blastocyst is listed in Appendix A. Individual blastocysts were thawed with Vit Kit-Thaw kits (FUJIFILM Irvine Scientific, Santa Ana, CA, USA, Cat# 90137-SO) according to the manufacturer’s instructions, and then incubated in acidic Tyrode’s solution to remove zona pellucidae, followed by three washes in 0.1% PBS/PVP buffer. Denuded blastocysts were then processed for various assays.

### 2.3. Genomic DNA and mRNA Separation from Individual Human Blastocysts

To isolate genomic DNA and mRNA, the individual denuded blastocyst was micro-pipetted into a PCR tube with 0.5 µL of 0.1% PBS/PVP buffer under a microscope to confirm no loss of the blastocyst. Then, 2.5 µL of RLT plus lysis buffer (Qiagen, Hilden, Germany, Cat# 1053393) was added into each PCR tube. Lysed blastocysts in PCR tubes were stored in a −80 °C freezer until further analysis. Genomic DNA and mRNA of each blastocyst were extracted simultaneously by following the G&T-seq protocol with modifications [22]. The oligos’ sequences and reagents used here can be found in the G&T-seq protocol. Briefly, 4 µL of Dynabeads per blastocyst was prepared for banding oligo-dT30VN, and then 50 µL of G&T-seq wash buffer was used to wash away DNA. Separated DNA was collected in a new PCR tube and stored in a −80 °C freezer. Isolated mRNA was transcribed to cDNA with 10 µL of reverse transcription (RT) master mix. Once the RT reaction was finished, the cDNA product was immediately amplified using the Kapa HiFi HotStart ReadyMix and IS PCR primer with a total volume of 15 µL per PCR reaction. Amplified cDNA was purified using AMPure X beads with 50 µL of water to elute, and then was stored in a −20 °C freezer for the gene expression assay.

### 2.4. Real-Time PCR for Relative Telomerase Gene Expression and Telomere Length Measurement

The purified cDNA of each blastocyst was diluted 50 times by adding nuclease-free water before the PCR setup. Each reaction contained 5 µL of diluted cDNA, 10 µL of SYBR green Supermix (Bio-Rad, Hercules, CA, USA, Cat#1708882), 1 µL each of forward and reverse primer (4 μM), and 3 μL of nuclease-free water. Duplicate reactions were set up for each sample per targetand then run through the program “CFX_3stepAMP”, with the annealing temperature set as 60 °C on the Bio-Rad CFX96 Real-Time System (Bio-Rad, Hercules, CA, USA). The melting curve was run to confirm the specific amplification for each pair of primers. A housekeeping gene, *GAPDH*, was run as an internal control. The relative telomerase *TERT* gene expression was compared using the 2^−ΔΔCt^ method.

Relative telomere length was measured by a qPCR method that Cawthon [23] developed in 2002, which was modified for the use of small amounts of DNA. A multi-copy gene, *5S rDNA* [24], served as an internal reference instead of the single copy gene *36B4* to avoid single copy gene amplification drop-off. A total of 20 µL of PCR reaction per primer pair was mixed by combining 10 µL of SYBR green Supermix, 0.5 µL each of forward and reverse primer (10 μM), 7 µL of nuclease-free water, and 2 μL of separated DNA after the G&T-seq protocol. Triplicate reactions per blastocyst per primer pair were set in a 96-well plate and then run through the program “CFX_2stepAMP”, with the annealing/extension temperature set as 60 °C. The melting curve analysis showed the single peak of the amplicon for each pair of primers. One euploid blastocyst was treated as the normalizer in different PCR plates, and relative telomere length was calculated as the T/S ratio by 2^−ΔΔCt^.

The primers for telomere length measurement and telomerase gene expression are listed in Table 1.

### 2.5. Immunofluorescence Staining 

Denuded blastocysts were fixed with 3.7% paraformaldehyde (PFA) in PBS for 10 min and then permeabilized in freshly made 0.5% Triton X-100 in 0.1% PVP/PBS for 15 min after rinsing 3 times in 0.1% PVP/PBS at room temperature. Fixed blastocysts were transferred into blocking buffer (2% goat serum, 1% BSA, 0.1% cold fish skin gelatin, 0.1% Triton X-100, 0.05% Tween 20, 0.05% sodium azide in PBS) and incubated overnight at 4 °C. Then, blastocysts were incubated in an anti-hTERT antibody (ThermoFisher Scientific, Waltham, MA, USA, Cat# PA5116024) solution diluted with blocking buffer in a 1 to 200 ratio at 37 °C for 1 h. Two aneuploid blastocysts were incubated with blocking buffer without the anti-hTERT antibody to serve as a negative control. All blastocysts, including the negative control blastocysts, were washed 3 times in blocking buffer for a total of 30 min, and then incubated with Goat anti-Rabbit IgG 2^nd^ antibody (ThermoFisher Scientific, Rockford, IL, USA, Cat#35552) diluted 1:400 in blocking buffer at 37 °C for 30 min. After the second antibody incubation, blastocysts were washed three times in blocking buffer for 15 min. Immunostained blastocysts were loaded on a pre-cleaned glass slide with 15 µL of VectorShield with DAPI (Vector Laboratories, Newark, CA, USA, Cat# H-1200-10) and sealed with a coverslip and clear nail polish. Images were taken with a Zeiss microscope equipped with epifluorescence optics.

### 2.6. Telomerase Activity Quantification in Individual Human Blastocysts by TRAP-qPCR Assay

The telomerase activity of individual human blastocysts was quantitatively compared using a commercial qPCR assay kit (ScienCell Research Laboratories, Carlsbad, CA, USA, Cat#8928) with minor modifications. Individual denuded blastocyst was placed in a PCR tube with 0.2 µL PBS/PVP after being washed three times in 0.1% PBS/PVP. Then, 1 µL of cell lysis buffer supplied with PMSF and β-mercaptoethanol, according to the manufacturer’s instructions, was added into the same PCR tube for 30 min at 4 °C. Lysed blastocysts in the PCR tubes were then flash-frozen in liquid nitrogen and stored in a −80 °C freezer. The telomerase reaction was set up in the same PCR tube containing blastocyst lysate by directly adding 4 µL of 5X telomerase reaction buffer and 15 µL of nuclease-free water. All PCR tubes containing the telomerase reaction mix were placed in a thermocycler to incubate at 37 °C for 3 h, followed by inactivation at 85 °C for 10 min. Real-time PCR reaction then was prepared with 2 µL of post telomerase reaction product, 2 µL TPS primer stock solution, 10 µL of 2X GoldNStart TaqGreen qPCR master mix, and 6 µL of nuclease-free water. The positive control and negative control supplied with the kit were prepared by following the instructions while preparing blastocysts. Each sample was set with a duplicate reaction in a 96-well PCR plate, and then run through the program in a Bio-Rad CFX96 thermocycler as per the manufacturer’s instructions. The relative telomerase activity of each blastocyst to the positive control was calculated by 2^−ΔCq^ (ΔCq = Cq of blastocyst − Cq of positive control).

## 3. Results

### 3.1. Evaluation of Telomere Length

Telomere length was measured by real-time PCR in 120 human blastocysts from 30 subjects. One blastocyst was excluded from data analysis due to failure of amplification. The relative telomere length of each blastocyst and its ploidy status are shown in Appendix A. Blastocysts were categorized into five groups according to chromosome status: (1) euploid (n = 4), (2) segmental/mosaic involving one chromosome (n = 6), (3) aneuploid involving one chromosome, either loss or gain (n = 35), (4) aneuploid involving two chromosomes (n = 25), and (5) aneuploid involving three or more chromosomes (complex abnormal) (n = 49). 

Telomere length did not significantly differ among the five groups (Kruskal–Wallis test, *p* = 0.258; Figure 1A). Mean telomere length in human blastocysts tended to increase with increasing chromosomal abnormalities, though linear regression did not show this trend to be statistically significant (R^2^ = 0.006, *p* = 0.36; Figure 1A). Telomere length in aneuploid blastocysts was markedly longer than that in euploid and segmental/mosaic embryos (2.238 ± 1.724 *vs.* 1.26 ± 0.349, *p* = 0.002 and 2.238 ± 1.724 *vs.* 1.752 ± 0.323, *p* = 0.029, respectively; Welch’s *t* test; Figure 1B). Telomeres in segmental/mosaic did not differ from those in euploid blastocysts (1.752 ± 0.323 *vs.* 1.26 ± 0.349; *t* test, *p* = 0.051; Figure 1B). In addition, aneuploid blastocysts presented greater heterogeneity in telomere length compared to euploid and mosaic/segmental blastocysts, as quantified by the F test to compare variances (F = 24.35, *p* = 0.02). 

To exclude the impact of patient-specific clinical parameters, e.g., age and BMI, we compared telomere length in sibling euploid and aneuploid embryos from three subjects (#26, #29 and #30; Appendix A). The telomere length in sibling aneuploid embryos did not differ from sibling euploid embryos (1.56 ± 0.60 *vs.* 1.26 ± 0.35; *t* test, *p* = 0.392; Figure 1C). 

Since aging is a major risk factor for aneuploidy, we sought to determine whether maternal age was associated with telomere length in aneuploid blastocysts. The scatter plots shown in Figure 1D indicate that telomere length in aneuploid blastocysts increased with advancing maternal age (R^2^ = 0.06, *p* = 0.01). Paternal age did not impact telomere length (R^2^ = 0.03, *p* = 0.09).

### 3.2. The Expression of the hTERT Gene

We isolated mRNA from the same blastocysts used to measure telomere length, and quantified the expression level of *TERT* mRNA, using the expression of *GAPDH* mRNA as a control. The levels of *TERT* mRNA in each blastocyst are shown in Appendix A. In concordance with telomere length comparison, the *TERT* mRNA level in aneuploid (median = 643.8; 95% CI [818.7, 1420]) was much higher than that in euploid blasts (median = 35; 95% CI [−512.7, 1050]) (Mann–Whitney test, *p* = 0.033; Figure 2A). *TERT* mRNA expression in segmental/mosaic did not differ from that in euploid or aneuploid blastocysts (Mann–Whitney test, *p* = 0.257 and *p* = 0.47, respectively; Figure 2A). Notably, significantly heterogeneous mRNA expression of *TERT* was observed in aneuploid compared to other blastocysts (Bartlett’s test, *p* < 0.01). Furthermore, we did not find a strong correlation between telomere length and *TERT* mRNA abundance (*r* = 0.103, *p =* 0.265; Figure 2B).

As *TERT* mRNA abundance does not always reflect the encoded TERT protein level, IF staining was performed on whole blastocysts to identify the location and expression of the TERT protein. Specific staining for the TERT protein was imaged in all tested blastocysts (Figure 3), whether euploid or aneuploid. Intriguingly, strong anti-hTERT staining foci appeared on condensed chromatin in dividing cells (Figure 3). These results suggest that telomerase genes are actively transcribed and translated in human embryos at the blastocyst stage, regardless of ploidy status. 

### 3.3. Comparison of Telomere Length and Telomerase Gene Expression in Aneuploid Blastocysts with Whole Chromosome Gain and Loss

Next, we analyzed whether telomere length or telomerase gene expression differed in aneuploid embryos exhibiting chromosome gain *vs*. loss. Telomere length and telomerase gene expression did not differ between 42 blastocysts with chromosome gain (≥47) and 35 blastocysts with chromosome loss (≤45) (*p* > 0.05, *t* test; Figure 4A). Telomere length and telomerase gene expression did not differ between trisomic (Trisomy 22, 20, 16, 11, 8, and 2) and monosomic blastocysts (Monosomy 22, 20, 16, 11, 8, and 2) (*p* > 0.05, *t* test; Figure 4B). Interestingly, aneuploid blastocysts with fewer chromosomes tended to have longer telomeres than those with more chromosomes (*p* = 0.072 in chromosome gain and loss and *p* = 0.075 in trisomy and monosomy; *t* test) (Table 2 and Figure 4). The parental age did not differ between blastocysts with chromosome loss *vs*. gain (Table 2).

### 3.4. Quantification of Telomerase Activity

Telomerase activity is directly involved in telomeric DNA synthesis, so we measured telomerase activity using qPCR in twenty aneuploid blastocysts and compared them to a telomerase positive control. All tested embryos had much smaller Cq Value compared to the inactivated telomerase positive sample (Appendix A), consistent with active telomerase in all human embryos at the blastocyst stage. Telomerase activity in euploid and mosaic blastocysts was significantly higher than in telomerase positive controls (6.36 ± 0.84 *vs.* 1.01 ± 0.12, *p* = 0.001 and 5.78 ± 2.39 *vs.* 1.01 ± 0.12, *p* = 0.02, respectively, by *t* test; Figure 5). Aneuploid blastocysts did not show significantly higher telomerase activity than the telomerase positive control (2.65 ± 1.40 *vs.* 1.01 ± 0.12, *p* = 0.058, *t* test; Figure 5). Strikingly, aneuploid blastocysts have decreased telomerase activity compared to euploid and mosaic blastocysts (2.65 ± 1.40 *vs.* 6.36 ± 0.84, *p* = 0.001 and 2.65 ± 1.40 *vs.* 5.78 ± 2.39, *p* = 0.001, respectively, by *t* test; Figure 5), though telomerase activity did not differ between mosaic and euploid blastocysts (6.36 ± 0.84 *vs.* 5.78 ± 2.39, *p* = 0.78; *t* test; Figure 5). These results suggest that telomerase remains active at the blastocyst stage, but at decreased levels in aneuploid blastocysts.

## 4. Discussion

Here, we report longer telomere length in aneuploid compared to euploid human blastocysts, even in sibling blastocysts. Unlike other studies [20,21] measuring telomere length in single cells or few cells from trophectoderm (TE) biopsy, we measured telomere length in whole embryos, which provide sufficient cells to avoid whole genomic amplification (WGA). Presumably, our data provide a more accurate estimate of telomere length, and involve less technical artifact. Moreover, differences may exist in telomere lengths between inner cell mass (ICM) and TE, limiting the generalizability of telomere length measurement in TE biopsies. For example, in mouse, the ICM has longer telomeres than TE, while in cattle the opposite is true [25,26]. Whether telomere length differs between ICM and TE in human blastocysts is unknown, but such differences may help to explain how our findings, carried out on whole embryos, conflict with previous studies on TE cells or blastomeres. 

Our data demonstrate that aneuploid blastocysts have longer telomeres but decreased telomerase activity compared to euploid or mosaic/segmental human blastocysts. The canonical function of telomerase is to maintain telomere ends by addition of the telomeric DNA repeat TTAGGG [27]. The lack of telomerase activity leads to progressive telomere erosion in dividing cells. In the present study, we analyzed the association of telomere length with telomerase gene expression in human blastocysts and found that longer telomeres in aneuploid blastocysts are accompanied by a higher level of telomerase *TERT* mRNA expression in aneuploid compared to euploid blastocysts, but lower telomerase activity. Studies demonstrate that telomerase gene expression does not always parallel telomerase activity [28,29]. The measurement of telomerase activity in individual blastocysts showed much lower telomerase activity in aneuploid compared with euploid blastocysts, which suggests that telomerase activity is down-regulated in aneuploid blastocysts. Intriguingly, aneuploid embryos maintain a robust telomere reserve despite diminished telomerase activity. We cannot rule out the possibility that robust telomere reserve may enable some aneuploid embryos to escape cellular senescence and death during preimplantation development. Previous studies [8,20,30] demonstrated that telomere length is reset between the cleavage and blastocyst stages of development. We previously demonstrated this robust (kilobase per cell cycle) telomere elongation during early mouse embryo development, at a stage when telomerase activity remains minimal [8], and this elongation is recombination-based alternative lengthening of telomeres (ALT). Telomere lengthening occurred even in a telomerase null mouse, suggesting that telomere length in embryos is set during the cleavage stage and only maintained by telomerase. Recombination-based ALT resets the short telomeres found in oocytes and zygotes, but at the cost of increased genomic instability which accompanies sister chromatid exchange. 

Our findings of increased telomerase gene expression and protein staining in the setting of decreased telomerase activity in aneuploid blastocysts raise the question of non-canonical functions for TERT during early development. A number of non-canonical functions of the telomerase catalytic component TERT have been reported in cultured cells, including the enhancement of cellular proliferation and survival, transcriptional regulation, protection from DNA damage, and regulation of mitochondrial function [5,31,32,33]. Intriguingly, condensed anti-hTERT staining foci localized to compressed chromatids in both euploid and aneuploid blastocysts, which suggests that TERT may be involved in mitotic cell division during human embryo development. This finding further supports telomerase-independent roles for TERT, as previously suggested [34]. Moreover, since telomere elongation is occurring in embryos with marked chromosome abnormalities, DNA damage, e.g., that resulting from aging, may promote TERT expression to enable these embryos to escape senescence [35,36]. Such phenomena and potential mechanisms merit further investigation.

The relative telomere length measured by qPCR is calculated by a ratio of the amount of telomere amplification product to that of a reference gene [37]. Therefore, extra chromosomes in aneuploid cells would be expected to increase not only the total amount of telomere but also the reference gene amplification product, so the ratio would remain unchanged. This may explain why our data show that average telomere length in embryos does not differ between embryos with a gain vs. loss of chromosome(s). It is well known that the expression of a substantial number of genes directly correlates with gene dose [38]; however, gene expression levels often do not reflect the actual gene copy number due to the dosage compensation and the global re-balancing of aneuploid genomes [39]. Here, we also found that elevated telomerase gene expression in aneuploid blastocysts was not affected by extra chromosome gain or loss. The TERT gene is located on chromosome 5, and our sample had only two embryos with a gain of chromosome 5 and none with a loss of chromosome 5 (Appendix A). Therefore, we cannot exclude the possibility that variation in the copy number of chromosome 5 could impact telomerase gene expression or telomere length.

These findings, for the first time, report that human blastocysts with single chromosome mosaicism or segmental abnormalities present indistinguishable telomere lengths, telomerase gene expression, and telomerase activity compared with euploid blastocysts. These findings indicate the similarity of developmental patterns between euploid and mosaic or segmental blastocysts, which could explain why mosaic and segmental embryo transfers result in healthy infants undergoing IVF treatment [40,41]. Hence, our work provides direct molecular evidence justifying the transfer of mosaic or segmental embryos in patients who have no other embryos.

## 5. Conclusions

Our study presents novel, quantitative data about telomere length, telomerase gene expression, and telomerase activity in human blastocysts. It demonstrates that there is a different landscape of telomere lengths and telomerase between euploid and aneuploid blastocysts, and it sheds light on the molecular mechanism underlying the survival of aneuploid blastocysts. Our findings are consistent with prior studies in mice showing that telomere length is reprogrammed by recombination at the early stages of embryonic development, and is maintained by telomerase during the later stages of preimplantation development, even in some aneuploid embryos. Robust TERT expression and telomere length maintenance in aneuploid human blastocysts may explain why extended in vitro culture alone is insufficient to cull out aneuploidy embryos during IVF. 

## Figures and Tables

**Figure 1 genes-14-01200-f001:**
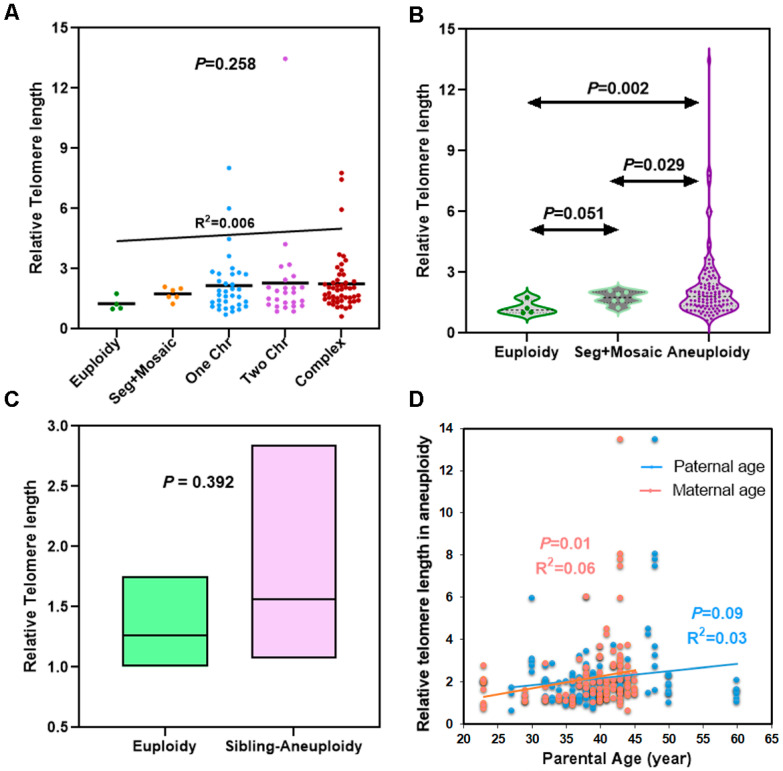
Relative telomere length in human blastocysts. (**A**) Comparison of average telomere lengths in human blastocysts grouped by the number of chromosomal errors. (**B**) Multiple comparison of average telomere length in aneuploidy, euploidy, and segmental/mosaic embryos. (**C**) Comparison of average telomere length between euploidy and sibling-aneuploidy from three subjects. Box bars depict from minimum to maximum with mean. (**D**) Association of telomere length in aneuploid blastocysts with parental age. The orange dots represent maternal age, and the blue dots represent paternal age.

**Figure 2 genes-14-01200-f002:**
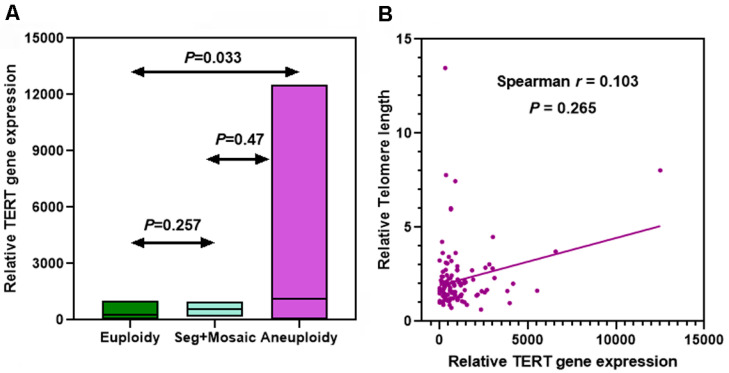
Telomerase *TERT* gene expression in human blastocysts by real-time PCR. (**A**) Comparison of *TERT* mRNA expression level. Box bars depict from minimum to maximum with median. (**B**) Association of telomere length with relative *TERT* gene expression by Spearman’s Rank Correlation analysis.

**Figure 3 genes-14-01200-f003:**
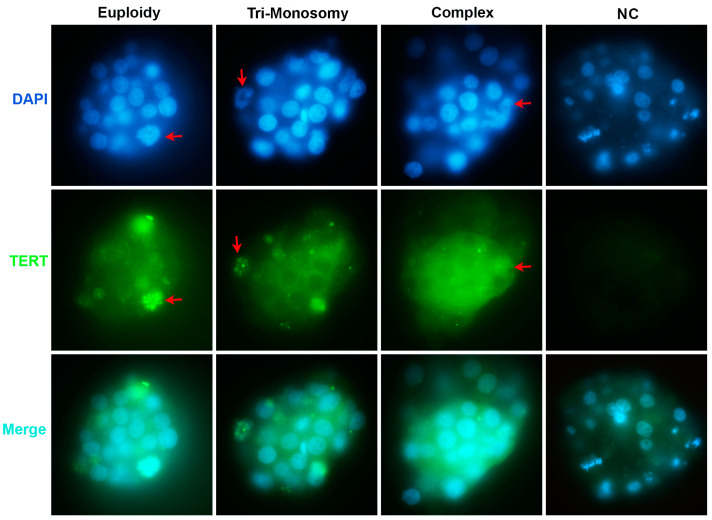
Immunofluorescent staining with anti-hTERT antibody in human blastocysts. NC represents negative control embryo stained only with secondary antibody. Red arrows point at the dividing cells with condensed anti-hTERT staining foci.

**Figure 4 genes-14-01200-f004:**
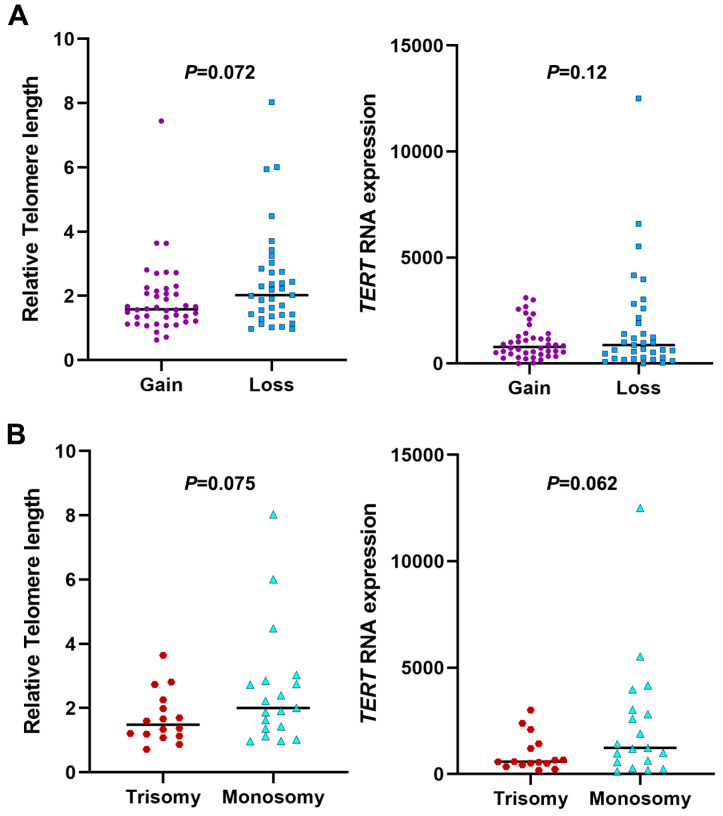
Whole chromosome gain did not increase telomere length and telomerase gene *TERT* expression in human aneuploid blastocysts. (**A**) Comparison of telomere length and *TERT* mRNA expression level in aneuploidy between whole chromosome gain (chromosome number ≥ 47, N = 42) and loss (chromosome number ≤ 45, N = 35). (**B**) Comparison of telomere length and *TERT* mRNA expression level in aneuploidy between trisomy (N = 9) and monosomy (N = 12).

**Figure 5 genes-14-01200-f005:**
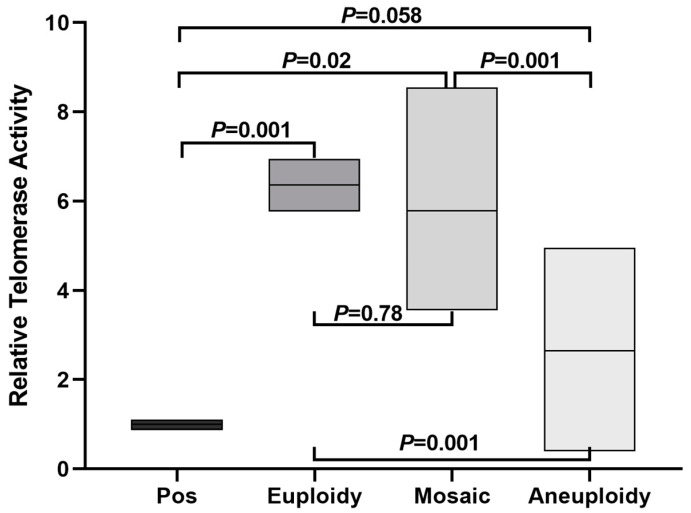
Comparison of relative telomere activity in human blastocysts. Pos represents telomerase positive control sample supplied by the assay kit. *p* value was calculated by *t* test on the GraphPad Prism 9 platform. The box bar represents from the minimum to maximum with mean value.

**Table 1 genes-14-01200-t001:** Sequences of primers for telomeres in the study.

Primer	Sequence
*GAPDH*-F	TTC ACC ACC ATG GAG AAG GC
*GAPDH*-R	CCC TTT TGG CTC CAC CCT
*TERT*-F	AAA TGC GGC CCC TGT TTC T
*TERT*-R	CAG TGC GTC TTG AGG AGC A
Telomere-F	CGG TTT GTT TGG GTT TGG GTT TGG GTT TGG GTT TGG GTT
Telomere-R	GGC TTG CCT TAC CCT TAC CCT TAC CCT TAC CCT TAC CCT
*5S rDNA*-F	CTC GTC TGA TCT CGG AAG CTA AG
*5S rDNA*-R	GCG GTC TCC CAT CCA AGT AC

**Table 2 genes-14-01200-t002:** Comparison of telomere length and telomerase gene expression in aneuploid blastocysts between chromosomal gain and loss.

	Chromosome Gain ≥ 47 (N = 42)	Chromosome Loss ≤ 45 (N = 35)	*p*	Trisomy (N = 9)	Monosomy (N = 12)	*p*
Maternal age (y) *(Median (Minimum, Maximum])*	41 (23, 45)	38 (23, 45)	0.53	41 (23, 44)	38 (23, 43)	0.37
Paternal age (y) *(Median (Minimum, Maximum))*	39 (27, 60)	39 (29, 60)	0.86	39 (29, 60)	36 (29, 60)	0.45
Telomere length *(Mean ± Std.)*	1.87 ± 1.12	2.46 ± 1.58	0.072	1.72 ± 0.84	1.88 ± 0.67	0.075
*TERT* mRNA *(Median (25%–75%Percentile))*	783 (448–1309)	879 (271–2160)	0.12	621 (504–1926)	971 (256–1997)	0.062

## Data Availability

Not applicable.

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
