# Peer review of "The Landscape of Telomere Length and Telomerase in Human Embryos at Blastocyst Stage"

_genes, 2023, doi:10.3390/genes14061200_

Round 1
Reviewer 1 Report
The paper describes quantitative data regarding telomere lengths, telomerase activity and gene expression in human blastocysts. The study was very well planned and is well presented. The significance of this study for UVFs is not really underlined and maybe should be done in order for the reader to understand the importance of the findings.
I have no remarks except for recommendation to emphasize more the importance of the findings. Just one small thing: at the results section 3.2 line # 200- please correct it to The expression of the hTERT gene and not as written.
Author Response
We appreciated the reviewer’s comments. In discussion section, we have discussed and emphasized our findings on longer telomeres and higher hTERT expression in aneuploid embryos. And we have corrected the Line#200 as suggested in the revised manuscript.
Reviewer 2 Report
The subject of this investigation is certainly interesting and has scientific significance. But there are certain minor points that you need to pay attention to.
1. The size of the control group (euploid blastocysts n=4) and aneuploid blastocysts (n=109) are not equal. Moreover, the length of the telomere in aneuploid blastocysts was very variable (figure 1B). Therefore, it is not correct to conclude that the telomere length in aneuploid blastocysts exceeds that in euploid blastocysts. Besides, in n sibling euploid and aneuploid embryos this trend is not observed.
2. It is currently known that early embryos adopt a telomerase-independent mechanism known as alternative lengthening of telomeres (ALT) to elongate telomeres and telomerase activity remains low in early cleavage embryos. Previously, it was shown that from the blastocyst stage onwards, telomerase only maintains the telomere length established by a recombination-based mechanism (Liu et al., 2007). Therefore, the analysis of telomerase activity in blastocysts may not reflect the actual state.
I recommend to revise the manuscript in accordance with the comments
Author Response
The present study involved in human preimplantation embryos, as only few patients donated their euploid embryos we have limited euploid embryos for research purpose. Moreover, our goal is to profile telomeres in human aneuploid blastocysts and discuss the role of telomeres in aneuploid blastocysts development. The conclusion mentioned in comment#1 was based on average telomere length comparison between aneuploid and euploid embryos, which means some individual aneuploid blastocyst have longer telomeres while some have shorter telomeres in comparison with euploid blastocysts. For instance, it is known that telomere length in older population is shorter than that in younger population, but many individual older persons could have longer telomeres. The results in the sibling euploid and aneuploid embryos showed the trend of longer telomere length in aneuploid embryos, but the sample size is too small to show the significance.
In discussion Line #301 - #308, our discussion agreed to the comment #2. Our analysis on telomerase activity presented different profile of lower telomerase activity in aneuploid blastocysts compared to euploid, and we didn’t correlated telomerase activity with telomere length in present study. Moreover, we have argued that telomerase TERT in aneuploid embryos may has non-canonical functions beside lengthening telomere, that includes enhancing aneuploid embryo survival and regulating other genes (see discussion Line #310 - #322).